# Alkyphenol Exposure Alters Steroidogenesis in Male Lizard *Podarcis siculus*

**DOI:** 10.3390/ani11041003

**Published:** 2021-04-02

**Authors:** Mariana Di Lorenzo, Aldo Mileo, Vincenza Laforgia, Maria De Falco, Luigi Rosati

**Affiliations:** 1Department of Biology, University of Naples ‘‘Federico II’’, 80126 Naples, Italy; mariana.dilorenzo@unina.it (M.D.L.); mileo.aldo@gmail.com (A.M.); vincenza.laforgia@unina.it (V.L.); luigi.rosati@unina.it (L.R.); 2National Institute of Biostructures and Biosystems (INBB), 00136 Rome, Italy; 3Center for Studies on Bioinspired Agro-environmental Technology (BAT Center), 80055 Portici, Italy

**Keywords:** sex hormones, endocrine disrupting chemicals, Nonylphenol, Octylphenol, testis, steroidogenesis

## Abstract

**Simple Summary:**

The non-stop release and increase in the ecosystem of toxic and polluting substances such as endocrine disrupting chemicals pose a threat to the survival of wildlife. Nonylphenol and Octylphenol are well known xenobiotics, with estrogen-like properties, widely used to optimize the manufacturing of different products, and due to their hydrophobicity and low solubility, they are persistent and ubiquitous in many environmental matrices. The goal of the present study was to investigate the effects of these compounds on the testis of the lizard *Podarcis siculus* during the reproductive period, focusing our attention on the steroidogenesis process. The lizard *P. siculus* has been chosen as animal model as it is usually used as a sentinel species and because the morphology and physiology of testis between lizards and mammals do not change, allowing a fine correlation of observed effects to human health. Obtained results showed that both substances used in this study, are able to alter testis histology and localization of key steroidogenic enzymes such as 3β-hydroxysteroid dehydrogenase, 17β-hydroxysteroid dehydrogenase and P450 aromatase, which represents the major target of these two alkylphenols. In conclusion, Nonylphenol and Octylphenol inhibit steroidogenesis, interfering with the reproductive capacity of the lizard *Podarcis siculus*.

**Abstract:**

Background: Nonylphenol (NP) and Octylphenol (OP) are persistent and non-biodegradable environmental contaminants classified as endocrine disruptor chemicals (EDCs). These compounds are widely used in several industrial applications and present estrogen-like properties, which have extensively been studied in aquatic organisms. The present study aimed to verify the interference of these compounds alone, and in mixture, on the reproductive cycle of the male terrestrial vertebrate *Podarcis siculus,* focusing mainly on the steroidogenesis process. Methods: Male lizards have been treated with different injections of both NP and OP alone and in mixture, and evaluation has been carried out using a histological approach. Results: Results obtained showed that both substances are able to alter both testis histology and localization of key steroidogenic enzymes, such as 3β-hydroxysteroid dehydrogenase (3β-HSD), 17β- hydroxysteroid dehydrogenase (17β-HSD) and P450 aromatase. Moreover, OP exerts a preponderant effect, and the P450 aromatase represents the major target of both chemicals. Conclusions: In conclusion, NP and OP inhibit steroidogenesis, which in turn may reduce the reproductive capacity of the specimens.

## 1. Introduction

In the last few decades, the survival of wildlife has been seriously threatened by the continuous and rapid increase in the process of urbanization and industrialization.

Notwithstanding the man-made products, which apparently represent a great benefit, they are not very compatible with the environment, and represent one of the main causes of the dissipation of natural resources and the non-stop release and increase into the ecosystem of toxic and polluting substances. Over the period 2011–2018, sales of chemicals belonging to the family of endocrine disruptors chemicals (EDCs) were seen to remain stable at around 360 million kg/year in the European Union (EU), and the main four agricultural countries, France, Spain, Italy and Germany, accounted for more than two thirds of these sales in the EU [1]. This evidence is particularly important as a high rate of sale is directly proportional to a greater use and consequently greater release of residues into the surrounding environment, resulting in a threat to the entire terrestrial ecosystem [2].

EDCs represent a large family of substances capable of interfering with the endocrine system at multiple levels and therefore, contribute to an increased risk of adverse health outcomes throughout life [3,4].

Nonylphenol (NP) and Octylphenol (OP) are well known EDCs widely used to optimize the manufacturing of several common products such as pesticides, herbicides, lubricants, plastics, detergents and cosmetics [5,6]. These compounds own high hydrophobicity and low solubility, thus, they are persistent and ubiquitous in many environmental matrices, such as aquatic ecosystems, agriculture soil and sediments [7,8], and they have also been detected in airborne particles and indoor air [6]. Moreover, they can easily enter the food chain as they have been found to accumulate in a variety of seafood and aquatic organisms [9,10], as well as having been detected in drinking water [11]. In addition, recent studies have pointed out the presence of traces of these substances in human blood, serum, breast milk, amniotic fluid and urine [12,13] indicating that human exposure is unavoidable [14].

Due to their affinity to bind to the estrogen receptors (ERs), even if it is lower compared to the endogenous hormone 17-b-estradiol (E2), their mode of action is thought to be mediated by an interaction with ERs [15,16,17,18], thus envisaging agonistic or antagonistic activity on the target tissues [19].

Accumulating evidence reported that the main target is represented by the male and female reproductive systems. In males, this can negatively affect sperm vitality and density and testicular cancer, as well as prostate cancer; in females, the exposure is associated with menstrual cycle alteration, sex ratio imbalance and breast cancer [15,16,20,21,22].

Moreover, alkylphenols (APs) are capable of interfering with the reproductive hormonal system in different organisms [23,24,25], and most of the studies have been conducted on aquatic vertebrates where it has been shown that APs, due to their estrogenic potency, cause adverse effects on both testis and ovaries with a consequent marked alteration in sexual development, including gonadal maturation, spawning time, egg and sperm production [26,27]. Nonylphenol exposure in female mice induces the early beginning of puberty and pathohistological abnormalities in the uterus and ovaries [28]; in male specimens, it causes disruption of Sertoli cells, oxidative stress, and alters sex hormone production, thereby disrupting spermatogenesis [29]. In addition, NP affects the spermatogenesis and the morpho-physiology of both testis and epididymis of the terrestrial vertebrate *Podarcis siculus* [30].

Thus, in light of the above data, the aim of the present study was to investigate the effects of NP and OP on the testis of the lizard *Podarcis siculus* during the reproductive period, focusing our attention on the localization of key steroidogenic enzymes involved in the endocrine regulation of the spermatogenic cycle and, considering that one concern about EDCs is their potential cocktail effects, compounds given have been tested alone, and in mixture.

*Podarcis siculus* presents a tubular testis organization with a seasonal reproductive cycle whereby gonads are active only when temperature and photoperiod are more favorable [31,32], and for this reason it is a powerful experimental model for studying reproduction [33,34,35,36,37,38,39,40]; moreover, it is widespread in Southern Italy inhabiting pristine areas, city parks and cultivated fields, and its ecological and life history features make it an excellent sentinel species for the biomonitoring of the ecotoxicological impact of EDCs [41,42].

## 2. Material and Methods

### 2.1. Chemicals

Nonylphenol (NP) and Octylphenol (OP) were obtained from FLUKA (Sigma-Aldrich Co., St. Louis, MO, USA).

### 2.2. Animals

Animal experiments were performed according to the ethical provisions imposed by the European Union and permitted by the National Committee of the Italian Ministry of Health on *in vivo* experimentation.

Male specimens of the lizard (*Podarcis siculus*) were captured during the reproductive period (May–June 2013) in Campania (Southern Italy; Latitude: 41°19′54″ N; Longitude: 13°59′29″ E) and precisely, in the area near the metropolitan city of Naples, which is the one with the highest population density in Europe.

Animals were placed in large soil-filled terraria containing heather and, indoors subjected to a photoperiod (11 h of daylight) and natural temperatures (15–24 °C); moreover, they were daily fed with live fly larvae *Tenebrio molitor*, water was available ad libitum, and to reverse capture-related stress conditions, an acclimatization period of 15 days was allowed before starting the treatments [43].

### 2.3. Treatments

NP and OP were dissolved in 50 µL of corn oil used, respectively, at concentrations of 0.172 µg and 0.161 µg based on previously studies [44,45,46,47] and administered through intraperitoneal injections every 2 days.

Lizards were divided into five groups each consisting of ten animals:

Group Control: lizards intraperitoneally injected with twelve (12) injections, each of 50 µL of corn oil and sacrificed 24 h after the last injection.

Group I: lizards treated with twelve (12) intraperitoneal injections of NP and sacrificed 24 h after the last injection.

Group II: lizards treated with twelve (12) intraperitoneal injections of OP and sacrificed 24 h after the last injection.

Group III: lizards treated with ten (10) intraperitoneal injections of NP + OP and sacrificed 24 h after the last injection.

Group IV: lizards treated with seventeen (17) intraperitoneal injections of NP + OP and sacrificed 24 h after the last injection.

Lizards were inspected daily for signs of toxicity and death, and the day after the last injection, were anesthetized and sacrificed by decapitation.

### 2.4. Histological and Immunohistochemistry Analysis

To evaluate testicular morphology, testes were fixed in Bouin for 24 h at room temperature, dehydrated in ascending alcohols, paraffin embedded, and cut into 5-μm serial sections using a rotative microtome. Subsequently, the sections were subjected to routine histological analysis and stained with hematoxylin and eosin (H&E). Morphometric evaluation was carried out measuring connective tissue height, germ cell area and testis cord diameters, which were measured as previously reported in Di Lorenzo et al. [48].

For immunohistochemistry, poly-_L_-lysine slides (Menzel-Glaser, Braunschweig, Germany) with 5-µm thick sections were dewaxed, rehydrated in a graded series of alcohol, heat-treated (microwave) for 20 min in 10 mM citrate (pH 6.0) antigen-retrieval buffer, and then washed in PBS [49,50]. Briefly, to reduce endogenous peroxidase activity, sections were treated with 2.5% H_2_O_2_ for 40 min at room temperature, and then blocked for 1 h at room temperature with normal goat serum (Pierce, Rockford, IL, USA).

Slides were then incubated overnight at 4 °C with the following primary antibodies diluted in normal goat serum:

Rabbit anti-human 3β-HSD (1:100, Santa Cruz Biotechnology, Santa Cruz, CA, USA);

Rabbit anti-mouse 17β-HSD (1:100, Santa Cruz Biotechnology, Santa Cruz, CA, USA);

Rabbit anti-human P450 aromatase (1:200, Santa Cruz Biotechnology, Santa Cruz, CA, USA).

These antibodies have been previously validated in *Podarcis siculus* as described in Rosati et al. [51]. The day after, sections were washed in PBS and incubated with HRP-conjugated goat-anti rabbit/mouse secondary antibody diluted 1:200 in normal goat serum for 1 h at room temperature. Finally, sections were stained using diaminobenzidine (DAB) as chromogen, and counterstained with Meyer’s hematoxylin. Negative controls were carried out by omitting incubation with primary antibody.

Histology slides were observed using a Zeiss Axioskop microscope and the images were acquired using Axiovision 4.7 Software (Zeiss).

### 2.5. Statistical Analysis

Statistical analysis was performed using GraphPad Prism 8 software. Data obtained were expressed as means ± standard error of mean (SEM). The statistical significance was calculated using one-way ANOVA followed by Bonferroni’s multiple comparison test, and differences were considered statistically significant when the *p* values were at least *p* < 0.05.

## 3. Results

### 3.1. NP and OP Induce Alteration in Testis Morphology

In the testis of the control group, the seminiferous tubules, as expected in the reproductive period, showed Sertoli cells and germ cells in all stages of the spermatogenic cycle from spermatogonia (Spg) to spermatozoa (Spz); moreover, Leydig cells (LC) were present in the interstitial spaces (Figure 1A). Furthermore, in the examples of control, the tubules are characterized by a large lumen and many layers of superimposed germ cells (Figure 2A,B).

In specimens treated with 12 I NP, it was still possible to highlight the presence of germ cells in all stages of differentiation, but only a few spermatozoa were detectable in the wide lumen of the tubules (Figure 1B). In addition, the treatment with 12 I NP induced, compared to the control group, a reduction of tubule diameter, an area of the connective tissue surrounding the tubules and thickness of the germinative epithelium (Figure 2A,B). Moreover, tubules also showed several evident empty spaces, as in accordance with a reduction in the total amount of germ cells. This reduction of germ cells was more prominent after treatment with 12 I OP which, in turn, caused a further increase in the area of the connective tissue (Figure 2A–C). In addition, in the tubules treated with 12I OP, spermatids and spermatozoa were absent (Figure 1C).

Morphological analysis of the testis in specimens treated with 10 injections of the mixture NP + OP showed a huge reduction in the area of connective tissue and in addition, a reduction in the diameter of the tubules, and in the thickness of the germinative epithelium (Figure 2A–C), which resulted composed by spermatogonia (Spg), spermatocytes I (Spc I) and II (Spc II), few spermatids (Spd), and no spermatozoa in the lumen (Figure 1D).

After 17 injections of NP + OP, testes displayed a strong reduction of both tubule diameters and germinative epithelium thickness and were made up of only spermatogonia (Spg) and spermatocytes (Spc I and II) (Figure 1E and Figure 2A,B). Interestingly, this dose did not seem to determine any change in the connective tissue area compared to the control group (Figure 2C).

Furthermore, all the treatments did not cause significant alterations in the Sertoli and Leydig cells, which appeared to be distributed in a similar manner in all experimental groups (Figure 1B–E).

### 3.2. Immunohistochemistry for 3 βHSD, 17 βHSD and P450 Aromatase

During the reproductive period in male lizard *Podarcis siculus*, immunopositivity for 3βHSD, 17 βHSD was recorded both in germ and somatic cells, as previously reported in Rosati et al. [51]. Differently, P450 localization was detected within Leydig cells, spermatidis and spermatozoa with a faint signal in spermatogonia and Sertoli cells, and an absent signal among spermatocytes I and II [38].

After 12 injections of NP, germ cells as spermatogonia and spermatocytes I and II, as well as Sertoli cells, showed a signal for 3βHSD; moreover, the immunohistochemical signal was strong in spermatids and spermatozoa, and in the cytoplasm of Leydig cells (Figure 3A,B). The immunopositive signal for 17βHSD was recorded in spermatogonia, in spermatocytes I and II, in the tail of spermatozoa, and in the cytoplasm of Leydig and Sertoli cells. A faint signal was labeled in spermatids (Figure 3C–E). The signal for P450 aromatase was dully highlighted in spermatogonia, spermatocytes I and II, and cytoplasm of Leydig cells. A strong immunohistochemical signal was found in spermatids and spermatozoa. No signal was detected in Sertoli cells (Figure 3C–E).

In testes treated with 12 injections of OP, 3βHSD showed markedly weaker staining in spermatogonia, spermatocytes I and II, and in Sertoli cells; no signal was detected in the cytoplasm of Leydig cells (Figure 4A,B, insert B). Similarly, no signal for 17 βHSD was found within Leydig cells (Figure 4C,D). Moreover, a signal for 17 βHSD was detected in mitotic and meiotic germ cells as spermatogonia and spermatocytes I and II, and in the cytoplasm of the somatic Sertoli cells (Figure 4C,D). P450 aromatase was weakly labeled only in spermatocytes I and II (Figure 4E–G).

After treatment with 10 injections of the mixture NP + OP, a weak signal for 3βHSD was evident in the cytoplasm of spermatocytes I and II, and a faint signal was also present in spermatogonia, spermatids, and Leydig and Sertoli cells (Figure 5A–C, insert C). Regarding 17 βHSD, a tenuous signal was revealed only in spermatogonia, spermatocytes I and II, and in the cytoplasm of Sertoli cells (Figure 5., D, E, insert E). A faint signal was evident in spermatids (Figure 5, insert E). No signal was found in Leydig cells (Figure 5E). Finally, P450 aromatase was weakly localized only at the level of spermatocytes I and the few spermatids (Figure 5F–H).

Overlapping results were obtained after treatment with 17 I NP + OP regarding 3βHSD localization. In detail, a signal was present in the cytoplasm of spermatocytes I and in the cytoplasm of Leydig cells. Meanwhile, a faint signal was evident in spermatogonia and spermatocytes II, as well as Sertoli cells (Figure 6A,B). In testes treated with 17 I NP + OP, the immunopositive signal for 17βHSD was recorded in germ cells as spermatogonia, spermatocytes I and II. A light signal was labeled also in the cytoplasm of Leydig and Sertoli cells (Figure 6C,D). Moreover, no signal was recorded for P450 aromatase neither at the level of germ cells, except spermatocytes I, nor at the level of somatic cells (Figure 6E,F).

No immunohistochemical signal was found in the negative control sections (Figure 3, insert F; Figure 4, insert E; Figure 5, insert H; Figure 6, insert E).

## 4. Discussion

Results described above highlight that alkylphenols administered during the reproductive period affect the morpho-physiology of testis of the seasonal breeder *Podarcis siculus*.

Spermatogenesis, in fact, is regulated by pituitary gonadotropins, environmental factors and local testicular factors, including testosterone and 17-β-estradiol [40,52,53,54]. Specifically, spermatogenesis is arrested when 17-β-estradiol titers are high and testosterone titers are low and conversely, it is resumed when 17-β-estradiol titers are low and testosterone titers are higher [39]; consequently, it is clear that the balance of intratesticular levels of both androgens and estrogens acts as an on/off switch for spermatogenesis control [31,39]. For this reason, the presence in *Podarcis* testis of the enzymes directly involved in the sex hormones production, as 3β HSD, 17β HSD and P450 aromatase, play a pivotal role in the control of spermatogenesis [39,51]. Thus, considering the xenoestrogenic activity of both NP and OP, the purpose of this study was to investigate the mechanisms by which these chemicals affect spermatogenesis.

We demonstrated that both AP alter testis histo-architecture with a more preponderant harmful effect of OP than NP, indeed, in the group treated with 12 I of OP, compared to samples treated with 12 I of NP, it was possible to detect the absence of the last stages of differentiation (spermatids and spermatozoa), and a reduction of both seminiferous tubule diameter and area of germinative epithelium, indicating a failure in replacement of cells from the basis of the tubules. In addition, we have registered also an increase of intratesticular connective tissue.

The gonadotoxicity of APs has been already demonstrated in some fishes [24,55,56]; moreover, recently it has been shown that a NP-diet could induce apoptosis in testicular germ cells [30] and previously, a similar effect has been shown following treatment with the fungicide methyl thiophane [57]. Particularly intricate and interesting were the results obtained following the treatment with the mixture of alkylphenols, where it was again possible to notice the preponderant action of OP. especially in the group treated with 17I NP + OP. Indeed, the treatment with 10 I NP + OP determined a similar effect found in the samples treated with NP 12I, because likewise, the samples had a reduction in both tubule lumen and area of germinative epithelium, though testis treated with 10I NP+ OP are missing only spermatozoa. Differently, the treatment with NP + OP 17 I has determined an effect similar but stronger, than the treatment with the OP 10 I, indicating that the greater amount of OP present in the mixture may have won in competition with NP. This data confirms that exposure to multiple endocrine disrupting chemicals with similar or different modes of action lead to “cocktail” effects, and the combined exposure can lead to additive and synergistic effects [58,59,60]. Morphological results were supported by the localization of key enzymes requested for the biosynthesis of testosterone and 17-β-estradiol such as 3βHSD, 17βHSD and P450 aromatase.

Recently it has been demonstrated that all these enzymes are differently expressed from both germ and somatic cells during the different phases of the reproductive cycle, and that especially P450 aromatase, achieves a pivotal role in local control of spermatogenesis, and in particular spermiohistogenesis [39].

NP and OP both alone and in mixture significantly alter these enzyme localizations during the reproductive period and in this case, it is possible also to note prominent OP effects. Treatments induce alternation in the pattern of localization of all three studied enzymes and the inhibition of both 3βHSD and 17βHSD indicate that the treatments cause an altered biosynthesis of testosterone [61].

The major effects were about P450 aromatase, which converts androgens into estrogens [62], and this is quite interesting as P450 aromatase represents a balance molecule between androgen and estrogens levels as its absence also in germ cells, indicates a failure in the proliferative and meiotic phases of gametogenesis, which consequently leads to a failure in spermiogenesis. Taken all together, these findings suggested that APs, especially OP, could determine a reduction of the enzymes involved in the production of sex hormones, and in particular 3 β-HSD and P450 aromatase, which in turn, alter the balance between androgen and estrogen levels, thus affecting spermatogenesis, especially the spermiohistogenesis. This data could explain also in the testes treated, especially with OP, the absence of germ cells in the last stages of differentiation as spermatids and spermatozoa.

## 5. Conclusions

In conclusion, NP and OP inhibit steroidogenesis, which in turn, may reduce the reproductive capacity of the specimens.

*Podarcis siculus* proves to be once again an excellent experimental model, not only for the study of the mechanisms that control spermatogenesis, but also for the study of the effects of environmental pollutants on gonads.

## Figures and Tables

**Figure 1 animals-11-01003-f001:**
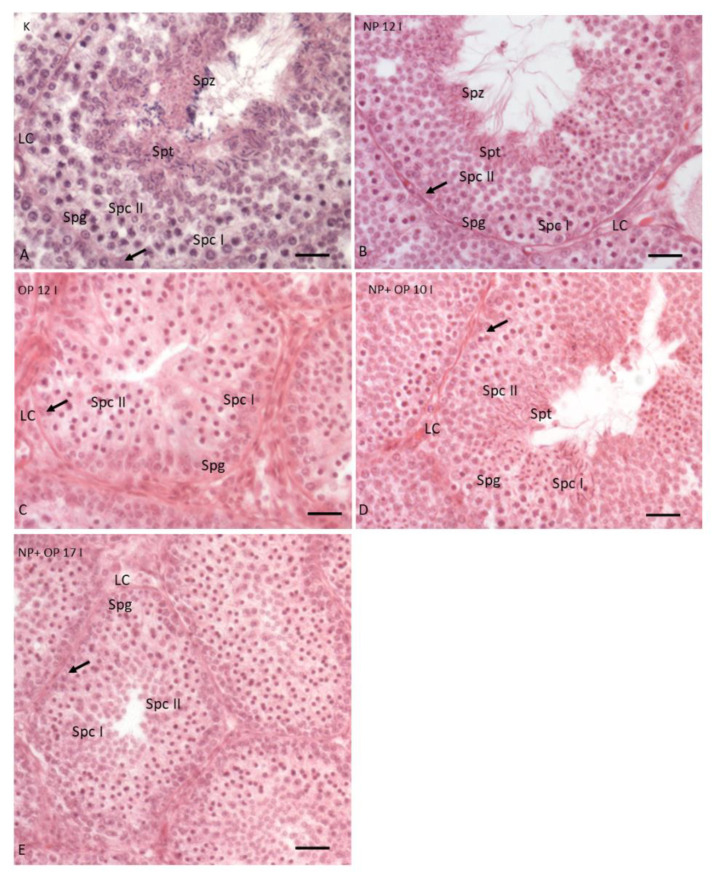
Testis from reproductive period. (**A**) Testis control. (**B**) Testis treated with 12I Nonylphenol (NP). (**C**) Testis treated with 12I Octylphenol (OP). (**D**) Testis treated with 10I NP + OP. (**E**) Testis treated with 17I NP + OP. Arrow: Sertoli cells; double arrowhead LC: Leydig cells, Spg: Spermatogonia, Spc I: Spermatocytes I, Spc II: Spermatocytes II, Spt: Spermatids, and Spz: Spermatozoa. Scale bars correspond to 20 µm in Figure 1A–E.

**Figure 2 animals-11-01003-f002:**
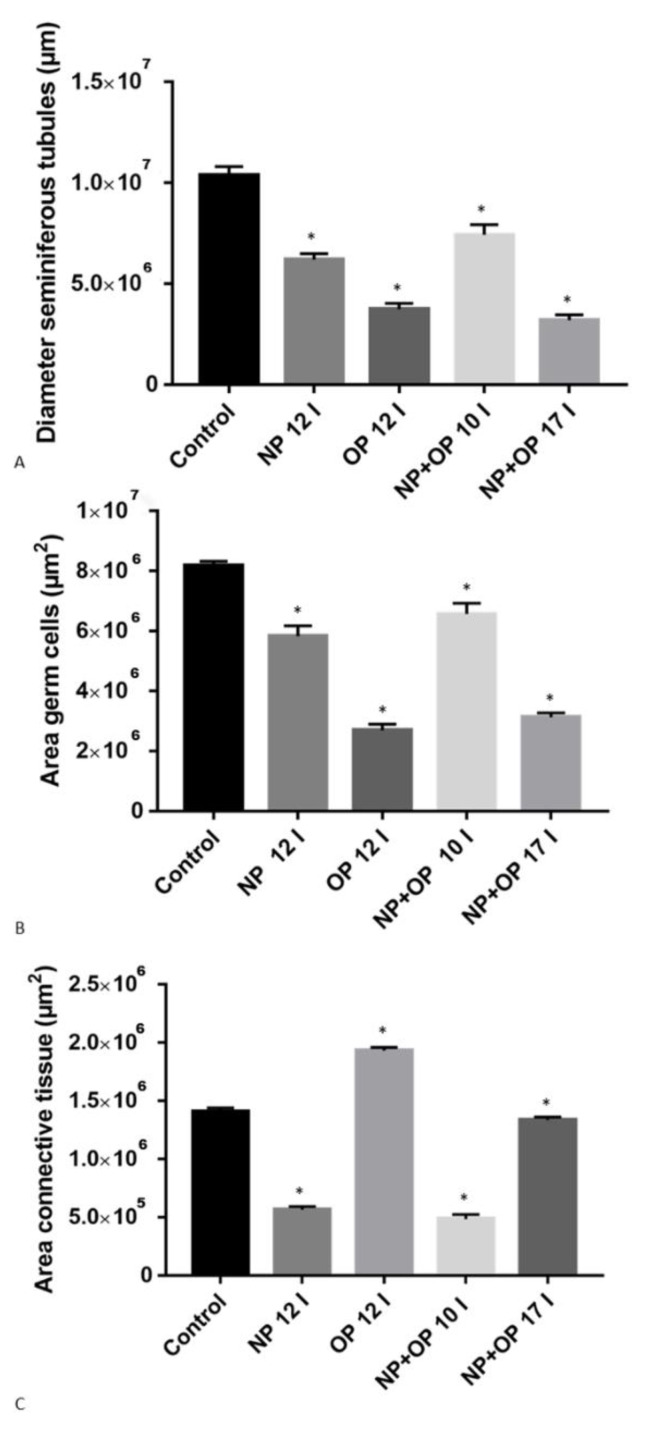
Morphometric analysis of *Podarcis siculus* testis treated with NP and OP during the reproductive period. (**A**) Diameter of the seminiferous tubules. (**B**) Area germ cells. (**C**) Area connective tissue. * *p* < 0.05.

**Figure 3 animals-11-01003-f003:**
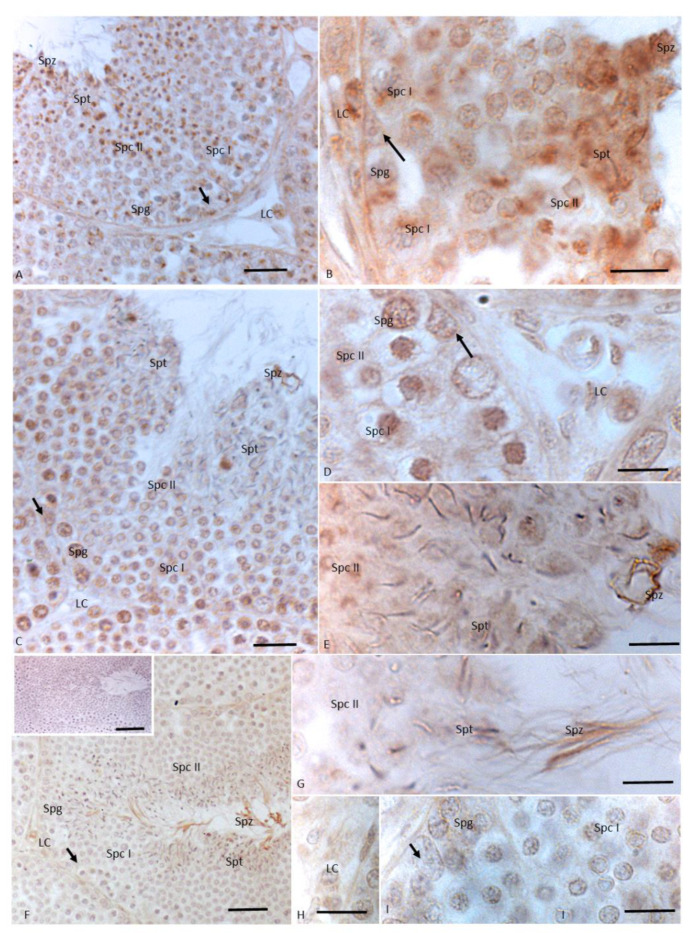
Immunohistochemical localization (brown areas) of 3βHSD, 17 βHSD and P450 aromatase in *Podarcis* testis treated with 12I NP. (**A**,**B**) 3βHSD distribution. Signal is evident in Sertoli (arrow) and Leydig (LC) cells, as well as in spermatogonia (Spg), spermatocytes I (Spc I) and II (Spc II), spermatids (Spt) and spermatozoa (Spz). (**C**–**E**) 17βHSD distribution. Signal is evident in Sertoli (arrow) and Leydig (LC) cells, as well as in spermatogonia (Spg), spermatocytes I (Spc I) and II (Spc II), spermatids (Spt) and spermatozoa (Spz). (**F**–**I)**) P450 aromatase distribution. Signal is evident in Leydig (LC) and germ cells as in spermatogonia (Spg), spermatocytes I (Spc I) and II (Spc II), spermatids (Spt) and spermatozoa (Spz). No signal is present in Sertoli cells (arrow). Insert (**F**) No signal is evident in control section. Scale bars correspond to 20 µm in (**A**,**C**,**F**) 5 µm in (**B**,**D**,**E**,**G**–**I**) and 50 µm in (**F**) insert.

**Figure 4 animals-11-01003-f004:**
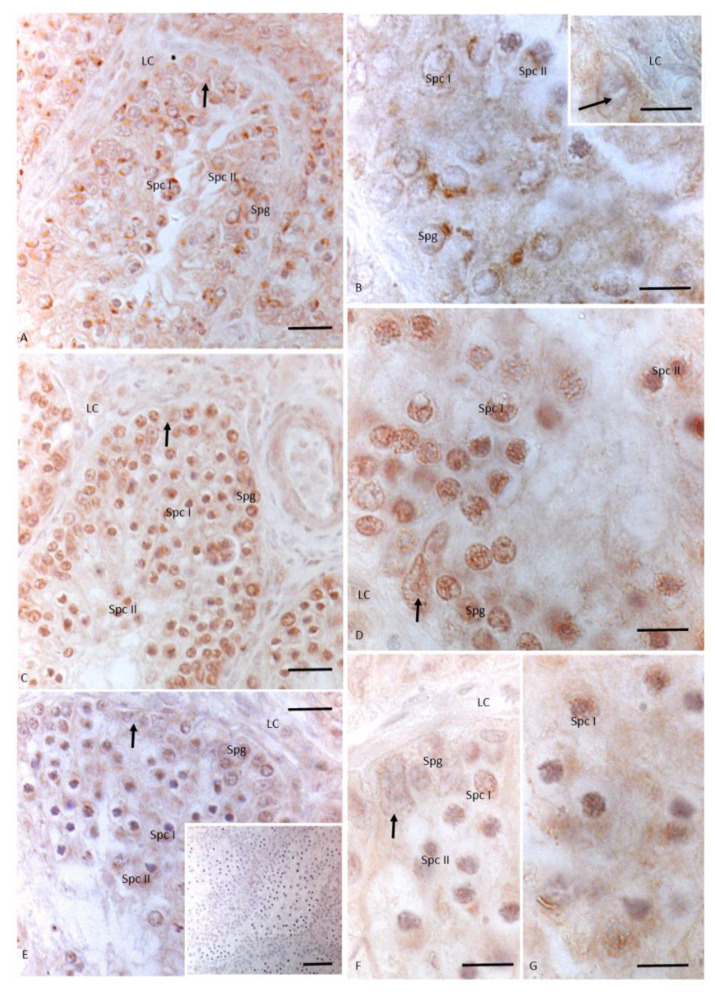
Immunohistochemical localization (brown areas) of 3βHSD, 17 βHSD and P450 aromaatse in Podarcis testis treated with 12I OP. (**A**,**B**) Insert (**B**) 3βHSD distribution. Signal is evident in Sertoli (arrow), in spermatogonia (Spg), spermatocytes I (Spc I) and II (Spc II). No signal is present in Leydig cells (LC). (**C**,**D**) 17βHSD distribution. Signal is evident in Sertoli (arrow), in spermatogonia (Spg), spermatocytes I (Spc I) and II (Spc II). No signal is present in Leydig cells (LC). (**E**–**G**) P450 aromatase distribution. Signal is weakly evident only in spermatocytes I and II. No signal is present in Leydig (LC) and Sertoli cells (arrow), as well as in spermatogonia (Spg). Insert (**E**) No signal is evident in control section. Scale bars correspond to 20 µm in (**A**,**C**,**E**) 5 µm in (**B**), (**B**) insert, (**D**,**F**,**G**) and 50 µm in insert (**E**).

**Figure 5 animals-11-01003-f005:**
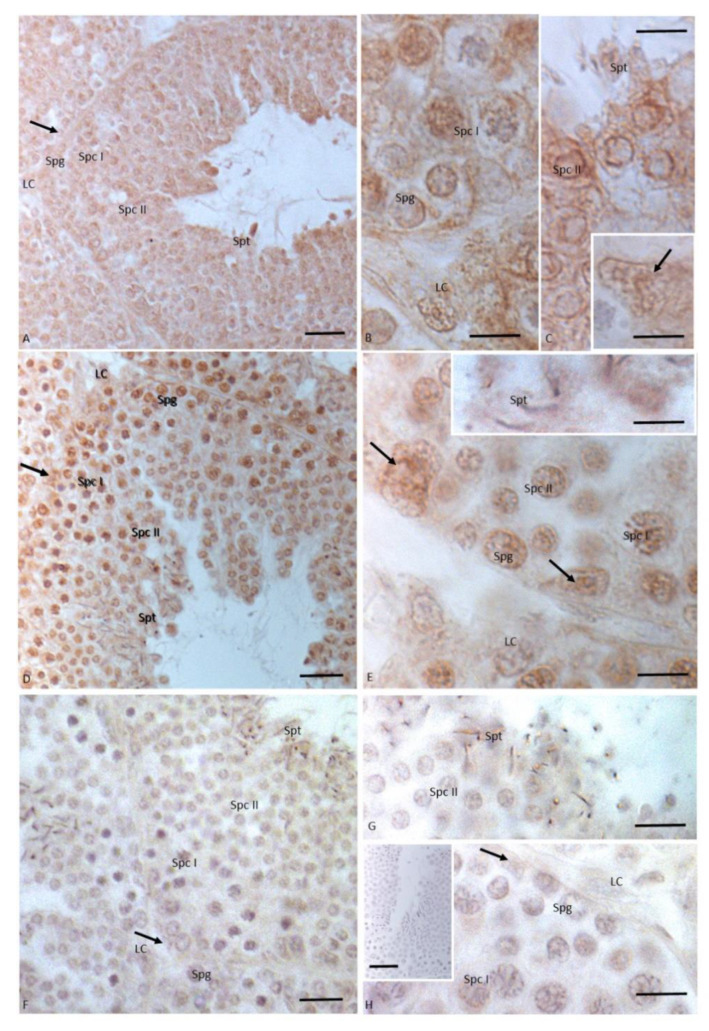
Immunohistochemical localization (brown areas) of 3βHSD, 17 βHSD and P450 aromatase in *Podarcis* testis treated with 10I NP + OP. (**A**–**C**) Insert (C) 3βHSD distribution. Signal is evident in Leydig (LC) and Sertoli cells (arrow), in spermatogonia (Spg), spermatocytes I (Spc I) and II (Spc II), and spermatids (Spt). (**D**,**E**)- insert E) 17βHSD distribution. Signal is evident in Sertoli (arrow), in spermatogonia (Spg), spermatocytes I (Spc I) and II (Spc II), spermatids (Spt). No signal is present in Leydig cells (LC). (**F**–**H**) P450 aromatase distribution. Signal is weakly evident only in spermatocytes I (Spc I) and spermatids (Spt). No signal is present in Leydig (LC) and Sertoli cells (arrow), as well as in spermatogonia (Spg), and spermatocytes II (Spc II). Insert (**H**): No signal is evident in control section. Scale bars correspond to 20 µm in (**A**,**D**,**F**) 5 µm in (**B**,**C**), insert (**C**,**E**), insert (**E**,**G**,**H**), and 50 µm in insert (H).

**Figure 6 animals-11-01003-f006:**
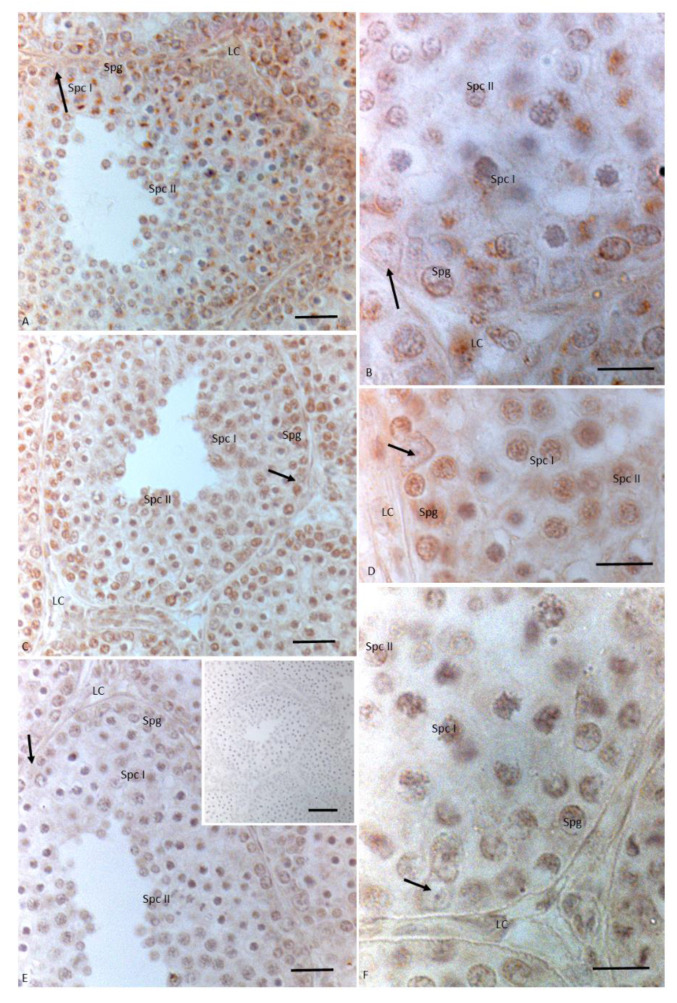
Immunohistochemical localization (brown areas) of 3βHSD, 17 βHSD and P450 aromatase in *Podarcis* testis treated with 17I NP + OP. (**A**,**B**)) 3βHSD distribution. Signal is evident in Leydig (LC) and Sertoli cells (arrow), in spermatogonia (Spg), spermatocytes I (Spc I) and II (Spc II). (**C**,**D**)) 17βHSD distribution. Signal is evident in Sertoli (arrow) and Leydig cells, in spermatogonia (Spg), spermatocytes I (Spc I) and II (Spc II). (**E**,**F**)) P450 aromatase distribution. Signal is weakly evident only in spermatocytes I (Spc I). No signal is present in Leydig (LC) cells and Sertoli cells (arrow), as well as in spermatogonia (Spg), and spermatocytes II (Spc II). Insert (**E**): No signal is evident in control section. Scale bars correspond to 20 µm in (**A**,**C**,**E**), 5 µm in (**B**,**D**,**F**) and 50 µm in insert (**E**).

## Data Availability

The data presented in this study are available on request from the corresponding author.

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
