# Peer review of "Alkyphenol Exposure Alters Steroidogenesis in Male Lizard Podarcis siculus"

_animals, 2021, doi:10.3390/ani11041003_

Round 1

Reviewer 1 Report

The manuscript is well prepared, however, it needs to be improved before publication.

Remarks:

  1. Abstract, line 22: "NP and OP could reduce steroidogenesis" the results of the experiment clearly proved that these EDCs inhibit testicullar steroidogenesis, therefore, better is to change this fragments such as: "In conclusion, NP and OP inhibit steroidogenesis which in turn may reduce the reproductive capacity of the specimens." The similar change should be done on page 13 (in Conclusions chapter).
  2. Introduction, page 2, lines 54-61: Here it is possible to find information concerning effects of NP and OP on female and male reproduction, however, these text fragment should be rearanged in oder to clearly show the influence of these ECDs separately on female and male reproduction (now everything is mixed).
  3. Introduction, page 2, lines 66-70: please put forward the hypothesis. Now, the main goal of the study is lack of data concerning effect of NP and OP on the reproductive system function in aquatic vertebrates.
  4. Material and methods: page 3, lines: 99-107: please explain why did you apply different numbers of NP, OP or NP+OP injections. Lines: 124-127: in the authors should provide the source of used antibodies.
  5. Fig. 1. Please increase the size of letters A, B, C D and E in the figure pictures. Now, it is not known which is picture A, B etc. Moreover, in this pictures descriptions shoud be written in larger characters.

Author Response

Reviewer 1

The manuscript is well prepared; however, it needs to be improved before publication.

Remarks:

  1. Abstract, line 22: "NP and OP could reduce steroidogenesis" the results of the experiment clearly proved that these EDCs inhibit testicular steroidogenesis, therefore, better is to change this fragments such as: "In conclusion, NP and OP inhibit steroidogenesis which in turn may reduce the reproductive capacity of the specimens." The similar change should be done on page 13 (in Conclusions chapter).

R: As suggested we have changed the sentences both in the abstract and in the conclusion.

  1. Introduction, page 2, lines 54-61: Here it is possible to find information concerning effects of NP and OP on female and male reproduction, however, these text fragment should be rearranged in order to clearly show the influence of these ECDs separately on female and male reproduction (now everything is mixed).

R: As suggested we have changed the sentence in Introduction section.

  1. Introduction, page 2, lines 66-70: please put forward the hypothesis. Now, the main goal of the study is lack of data concerning effect of NP and OP on the reproductive system function in aquatic vertebrates.

R: Thank you for your suggestion, in this section we have improved data about aquatic and terrestrial vertebrates.

  1. Material and methods: page 3, lines: 99-107: please explain why you applied different numbers of NP, OP or NP+OP injections. Lines: 124-127: in the authors should provide the source of used antibodies.

R: The concentrations used have been chosen in consideration of LD50 and according to preliminary dose response test and data. Moreover, as requested we have added info about source of antibodies.

  1. Fig. 1. Please increase the size of letters A, B, C D and E in the figure pictures. Now, it is not known which is picture A, B etc. Moreover, in these picture descriptions should be written in larger characters.

R: according to your suggested we have improved the size of letters.

Reviewer 2 Report

The present manuscript explores the effect of two alkylphenols in the testicular histology and steroidogenesis of the male lizard P siculus.

Introduction:

It is not clear why the authors choose this model and not another vertebrate?

Why did they made the experiments in the reproductive season?

Why did they work with males and not females?

Some info in the discussion (e.g. lines273-275) would fit better in the introduction.

Authors claim a wide gap in knowledge on vertebrates. There are several papers, that were not cited in this work,

Histological Analysis of Reproductive System in Low-Dose Nonylphenol-treated F1 Female Mice.

Kim YB, Cheon YP, Choi D, Lee SH.Dev Reprod. 2020 Sep;24(3):159-165. doi: 10.12717/DR.2020.24.3.159. Epub 2020 Sep 30.PMID: 331109   Effect of nonylphenol on spermatogenesis: A systematic review.

Malmir M, Faraji T, Ghafarizadeh AA, Khodabandelo H.Andrologia. 2020 Nov;52(10):e13748. doi: 10.1111/and.13748. Epub 2020 Jul 14.PMID: 32662580 Review.  

Methods:

What was the rationale to pick up the doses they used in this study?

I.P. injection is not the "natural way of exposure. Why they used this model instead of drinking water?

It is not clear the total days of treatment. Was it 24 days? Why this number of days. Which was the rationale

Why some groups were treated with 12 injections while others with 10 or 17?

it is not clear whether group's II and IV were treated with the same concentrations of NP and OP?

Discussion

Authors (lines 66-67) state they want to fill the gap in knowledge about EDC and its potential additive or synergic effect in mixtures. However, they do not make any mention of this in the discussion section. Even more, if they really want to explore these effect, the experimental design is wrong

Author Response

Reviewer 2

The present manuscript explores the effect of two alkylphenols in the testicular histology and steroidogenesis of the male lizard P. siculus.

Introduction:

It is not clear why the authors choose this model and not another vertebrate?

R: We used Podarcis siculus as animal model because it is usually used as bioindicator to relate the effects to human health and because essentially the morphology and physiology of the testis between lizards and mammals do not change.

Why did they make the experiments in the reproductive season?

R: We perform the experiments during the reproductive season because it is the period in which estrogen levels are lower and thus it is the best moment to investigate the potential effects of estrogen like compounds.

Why did they work with males and not females?

R: We chose males because in these, estrogen levels are lower, and a purpose of the study was to the harmful effects of estrogen like compounds on steroidogenesis in males.

Some info in the discussion (e.g. lines273-275) would fit better in the introduction.

R: Thank you for your suggestion, as requested we have moved lines 273-275 to the introduction section.

Authors claim a wide gap in knowledge on vertebrates. There are several papers, that were not cited in this work,

Histological Analysis of Reproductive System in Low-Dose Nonylphenol-treated F1 Female Mice.

Kim YB, Cheon YP, Choi D, Lee SH.Dev Reprod. 2020 Sep;24(3):159-165. doi: 10.12717/DR.2020.24.3.159. Epub 2020 Sep 30.PMID: 331109 Effect of nonylphenol on spermatogenesis: A systematic review.

Malmir M, Faraji T, Ghafarizadeh AA, Khodabandelo H.Andrologia. 2020 Nov;52(10):e13748. doi: 10.1111/and.13748. Epub 2020 Jul 14.PMID: 32662580 Review.

R: We apologize for the oversight, as suggested we have added data concerning these papers.

Methods:

What was the rationale to pick up the doses they used in this study?

R: The concentrations used have been chosen in consideration of LD50 and according to preliminary dose response test and data.

I.P. injection is not the "natural way of exposure. Why they used this model instead of drinking water? R: To be sure that each lizard receives the selected dose.

It is not clear the total days of treatment. Was it 24 days? Why this number of days? Which was the rationale?

R: The treatments lasted 20 days for the animals which have received 10 injections, 24 days for the animals which have received 12 injections and 34 days the animals which have received 17 injections. The number of the injections is strictly associated with the concentration of each compound which has been chosen in consideration of LD50 and time curse experiment.

Why some groups were treated with 12 injections while others with 10 or 17?

R: For the mixture groups we have showed data about the lowest and highest of injections capable of toxic effects compatible with LD50 of each compound.

It is not clear whether group's II and IV were treated with the same concentrations of NP and OP?

R: NP and OP, have been used at the concentration of 0.172 µg and 0.161 µg respectively specifically group II has received 12 injections of NP whereases group IV received 17 injections of NP and OP.

Discussion

Authors (lines 66-67) state they want to fill the gap in knowledge about EDC and its potential additive or synergic effect in mixtures. However, they do not make any mention of this in the

discussion section. Even more, if they really want to explore these effects, the experimental design is wrong.

R: Thank you for your suggestion, we have better expressed our aim of paper.

Round 2

Reviewer 2 Report

I have no further comments